environmental chemistry/analytical chemistry/ spectroscopy

dispersive liquid–liquid microextraction, magnetic nanoparticles, chloramphenicol, water samples, spectrophotometry

**Author for correspondence:**
Noorfatimah Yahaya
e-mail: noorfatimah@usm.my

# Magnetic nanoparticles assisted dispersive liquid–liquid microextraction of chloramphenicol in water samples

Salwani Md Saad[1], Nur Afiqah Aling[2], Mazidatulakmam Miskam[2], Mardiana Saaid[2], Nur Nadhirah Mohamad Zain[1], Sazlinda Kamaruzaman[3], Muggundha Raoov[4], Nor Suhaila Mohamad Hanapi[5], Wan Nazihah Wan Ibrahim[5] and Noorfatimah Yahaya[1]

[1]Integrative Medicine Cluster, Advanced Medical and Dental Institute (AMDI), Universiti Sains Malaysia, 13200 Bertam Kepala Batas, Penang, Malaysia
[2]School of Chemical Sciences, Universiti Sains Malaysia, 11800 Penang, Malaysia
[3]Department of Chemistry, Faculty of Science, Universiti Putra Malaysia, Serdang, Selangor, Malaysia
[4]University of Malaya Centre for Ionic Liquids (UMCIL), Department of Chemistry, Faculty of Science, Universiti Malaya, Kuala Lumpur 50603, Malaysia
[5]Faculty of Applied Sciences, Universiti Teknologi MARA, 40450 Shah Alam, Selangor, Malaysia

SMS, 0000-0002-7774-2956; MM, 0000-0001-9757-0883;
SK, 0000-0001-6299-8767; MR, 0000-0003-0304-0617;
NY, 0000-0002-3079-7837

This work describes the development of a new methodology based on magnetic nanoparticles assisted dispersive liquid–liquid microextraction (DLLME-MNPs) for preconcentration and extraction of chloramphenicol (CAP) antibiotic residues in water. The approach is based on the use of decanoic acid as the extraction solvent followed by the application of MNPs to magnetically retrieve the extraction solvent containing the extracted CAP. The coated MNPs were then desorbed with methanol, and the clean extract was analysed using ultraviolet–visible spectrophotometry. Several important parameters, such as the amount of decanoic acid, extraction time, stirring rate, amount of MNPs, type of desorption solvent, salt addition and sample pH, were evaluated and optimized. Optimum parameters were as follows: amount of decanoic acid: 200 mg; extraction time: 10 min; stirring rate: 800 rpm; amount of MNPs: 60 mg; desorption solvent: methanol; salt: 10%; and sample pH, 8. Under the optimum

conditions, the method demonstrated acceptable linearity ($R^2 = 0.9933$) over a concentration range of 50–1000 µg l$^{-1}$. Limit of detection and limit of quantification were 16.5 and 50.0 µg l$^{-1}$, respectively. Good analyte recovery (91–92.7%) and acceptable precision with good relative standard deviations (0.45–6.29%, $n = 3$) were obtained. The method was successfully applied to tap water and lake water samples. The proposed method is rapid, simple, reliable and environmentally friendly for the detection of CAP.

## 1. Introduction

Chloramphenicol (CAP) is a broad-spectrum antibiotic that often is administered to humans and animals owing to its effectiveness against Gram-positive and Gram-negative bacteria [1]. It has effective bactericidal activity against *Streptococcus pneumoniae*, *Neisseria meningitides* and *Haemophilus influenzae*, and it has bacteriostatic activity against most pathogens [2]. Unfortunately, CAP is also associated with deadly side effects such as bone marrow suppression, leukaemia and aplastic anaemia [3]. Hence, its usage is strictly controlled. Use of CAP in livestock production is prohibited in several countries, including China, the European Union and the United States [1]. Nevertheless, owing to its low price and antibiotic effectiveness, the illegal use of CAP in the livestock industry still occurs [1]. The pollution of CAP in environment water systems has attracted considerable attention and special environmental relevance owing to its difficult degradability in the metabolite system [4]. As a stable compound, CAP has a high leaching rate and is therefore easily excreted from agricultural waste into natural water [5].

CAP levels have been measured in surface water, groundwater and drinking water [6–8]. In Germany, 0.56 µg l$^{-1}$ of CAP was detected in sewage treatment plant effluents and 0.06 µg l$^{-1}$ in river water [9,10]. While there are reports of the presence of CAP in water matrices, to the best of our knowledge, no regulation has been established for CAP in water samples. Thus, sensitive and reliable detection techniques are crucial to regularly monitor the residual CAP present in environmental waters and to assist in the establishment of the maximum residue limits for CAP.

Measurement of CAP from various sample matrices has mostly been performed chromatographically using liquid chromatography [1,11–13] or gas chromatography [14,15]. Alternatively, ultraviolet–visible (UV–Vis) spectrophotometry can be performed to quantitate various compounds, such as biological macromolecules [16], transition metal ions [17], and highly conjugated organic compounds [18], with the added advantages of versatility, speed, easy operation, availability in routine laboratories, and economical operation costs. As direct detection of CAP is challenging owing to its minute levels in environmental water matrices, a highly sensitive, accurate and reliable sample preparation strategy is required to extract and concentrate CAP and to eliminate major interferences prior to analysis. Furthermore, an effective sample preparation step is required before spectroscopic analysis to increase sensitivity and selectivity.

Liquid–liquid extraction [1] and solid phase extraction [17] are commonly performed for this purpose. However, most of the developed classical separation approaches are relatively expensive, consume high volumes of organic solvents and are time consuming [1,17,18]. Owing to the drawbacks of conventional sample preparation methods, solvent-free extraction techniques or those using lower amounts of organic solvents are becoming more and more critical [19].

Dispersive liquid–liquid microextraction (DLLME) is an interesting approach that was developed by Assadi and co-workers in 2006 [20]. DLLME offers several advantages over conventional techniques, including rapid extraction, high efficiency, simple operation, low cost and no requirement of large amounts of extraction solvent. In DLLME, a cloudy solution is formed after rapid injection of a suitable mixture of extraction and disperser solvent into an aqueous sample by syringe. Owing to the large superficial area of contact between the phases, targeted analytes in the aqueous sample are rapidly extracted into the micro-droplets of the extraction solvent. The dispersion of the solvent in classical DLLME is commonly assisted by external chemical compound or by additional energy source such as mechanical stirring [21], ultrasound [22], heating [23] and microwave [24]. The disperser solvent may involve the partition distribution of the analytes because it increases their solubility in the sample solution and therefore it can reduce the potential efficacy of the technique [25]. Finally, after extraction, recovery of the solvent of phase separation usually requires a centrifugation step [26], which may slow down the overall extraction procedure [25]. Other alternatives have been proposed to simplify and speed up the recovery procedure. A previously reported work proposed the use of magnetic nanoparticles (MNPs) to improve the classic DLLME performance [27]. After extraction by the conventional DLLME step, MNPs are dispersed into the sample solution interacting *via* hydrogen bonding with organic

solvent. In a final step, the solvent–adsorbent combination is recovered using an external magnet field. A similar strategy has been developed for the extraction of nonylphenol on waters [28].

The goal of this study was to develop a simple, rapid, inexpensive and efficient sample preparation strategy for the measurement of CAP in water samples by coupling DLLME-MNPs and UV–Vis spectrophotometry. Heating and vortex steps were selected as the dispersive forces for the extractant (decanoic acid) and the MNPs required for its recovery, avoiding the employment of complicated external apparatus. To the best of our knowledge, this is the first report describing the application of DLLME-MNPs coupled with UV–Vis spectrophotometry to determine the CAP content in water samples.

# 2. Material and methods

## 2.1. Chemicals and reagents

High performance liquid chromatography (HPLC)-grade methanol and acetonitrile were purchased from Friendemann Schmidt (Parkwood, Western Australia). Ethanol, isopropanol and 1-propanol were obtained from QReC™ (Auckland, New Zealand). Decanoic acid was purchased from Merck (Darmstadt, Germany), and MNPs (Iron (II, III) oxide nanoparticles, ($Fe_3O_4$)) were supplied by Bendosen Laboratory Chemicals (Bendosen, Norway). CAP standard was purchased from Dr Ehrenstorfer GmbH (Augsburg, Germany). Sodium chloride (NaCl) and sodium hydroxide (NaOH) were purchased from R & M Chemicals (Essex, UK). All chemicals were of at least analytical reagent grade and were used without further purification. Ultrapure deionized water (resistivity, 18.2 MΩ cm$^{-1}$) was generated by a model Sartorius Milli-Q system (Göttingen, Germany). Stock solution of CAP (1000 mg l$^{-1}$) was prepared by dissolving an appropriate amount of CAP in acetonitrile. Working standard solutions were freshly prepared by diluting the stock solution with water.

## 2.2. Instruments

The absorbance of CAP was measured using a Perkin Elmer Precisely, Lambda 25 UV-Vis spectrometer (Waltham, MA, USA) at wavelength of 301 nm. A hotplate stirrer (Favorit, PLT Scientific Sdn. Bhd., Selangor, Malaysia) was used to mix the samples. The pH meter used throughout the experiment was a pH700 pH meter purchased from Eutech Instruments (Ayer Rajah Crescent, Singapore). A VIBRAX-VXR vortex mixer (IKA, Staufen, Germany) was used for vortex mixing, and a Branson 3510 Ultrasonicator (Danbury, CT, USA) was used for the extraction procedure. Structure and morphology of MNPs were characterized by field emission scanning electron microscopy (FESEM) (Hitachi, S-5200, Tokyo, Japan). Fourier-transform infrared (FTIR) spectroscopy was conducted on a PerkinElmer (2000 series) spectrometer (Waltham, Massachusetts, USA) using the KBr wafer technique and spectra were recorded in the range of 400–4000 cm$^{-1}$.

## 2.3. Preparation of sample solutions

Tap water was collected from the laboratory water tap at the School of Chemical Sciences, Universiti Sains Malaysia (Penang, Malaysia), and lake water was sampled from Tasik Fajar Harapan (Universiti Sains Malaysia, Penang, Malaysia). The water samples were filtered through a 0.45 µm nylon membrane filter (Whatman, Dassel, Germany), and pH was adjusted to pH 8 with 0.1 M NaOH prior to the DLLME-MNPs procedure.

## 2.4. Optimization of variables affecting the magnetic nanoparticles assisted dispersive liquid–liquid microextraction procedure

Several important parameters, including amount of decanoic acid, extraction time, stirring rate, amount of MNPs, type of desorption solvent, salt addition and sample pH were thoroughly optimized in this study. Approximately 20 ml of working solution containing 1 mg l$^{-1}$ of CAP was used for optimization purposes, and all experiments were repeated three times. Once the optimized conditions were identified, environmental water samples were analysed using the DLLME-MNPs extraction procedure.

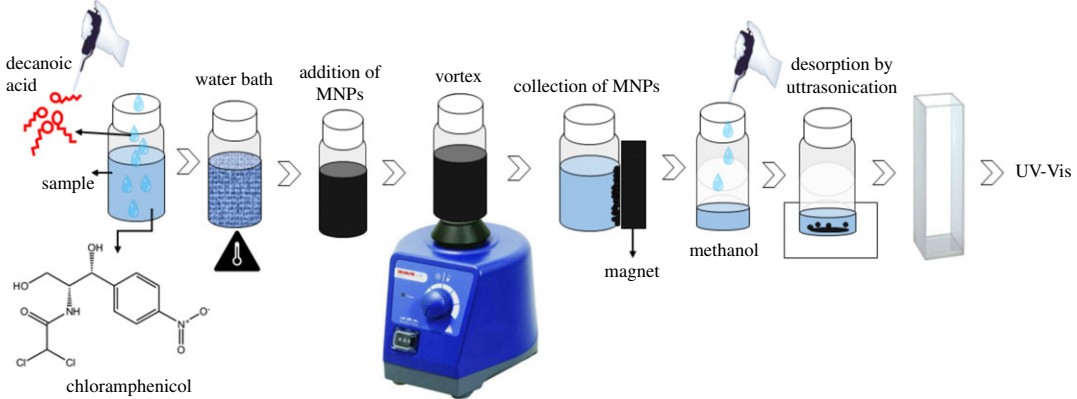

**Figure 1.** Schematic of the DLLME-MNPs procedure.

## 2.5. Magnetic nanoparticles assisted dispersive liquid–liquid microextraction procedure

For analysis of each water sample, a 20 ml aliquot of sample solution (pH 8) was added to a 40 ml sample vial. Next, 200 mg of decanoic acid was added to the sample solution as the extraction solvent. The mixture was agitated at 36°C in a water bath (Memmert GmBH, Schwabach, Germany). Following a 10 min incubation, 60 mg of MNPs were added to the sample solution, and the sample was vortexed (800 rpm) for 1 min. The MNPs were collected using an external magnet, and the supernatant was discarded. Next, 150 µl of methanol were added to desorb CAP from the MNPs surface. The mixture was sonicated for 2 min. An external magnet was applied, and the eluent was collected. The collected eluent was diluted with 850 µl of methanol and 500 µl of ultrapure water. Finally, the extracted samples were analysed using UV–Vis spectrophotometry. Figure 1 illustrates the extraction procedure.

## 2.6. Method validation

The developed DLLME-MNPs method was validated for linearity, limit of detection (LOD), limit of quantification (LOQ), repeatability and recovery under the optimized conditions. A seven-point calibration curve was plotted for CAP using water spiked with known concentrations of CAP. The experiments were carried out under the optimum conditions for each experimental parameter, and the absorbance of CAP was plotted against the concentration of CAP. CAP-free blank water samples were used for the validation study. LOD and LOQ were measured based on signal-to-noise ratio of 3 and 10, respectively. The repeatability was obtained from three replicates and expressed as relative standard deviations (RSDs) at low and high concentrations (50 and 200 µg l$^{-1}$) of CAP. The recoveries of all samples were calculated by comparing the final amounts of CAP after extraction from water samples with the corresponding spiking amounts on the calibration curve.

# 3. Results and discussion

## 3.1. Characterization of magnetic nanoparticles

Vibrating sample magnetometer analysis of MNPs were reported previously by our group [29]. Based on the FTIR spectrum (figure 2a), the bond at 579 cm$^{-1}$ in MNPs before extraction and after extraction refer to the stretch of Fe–O. A typical major absorption at 1709 cm$^{-1}$ indicated the absorption of COOH group of decanoic acid. Compared with the FTIR spectrum of bare MNPs (before extraction), the absorption signals at 2925 and 1412 cm$^{-1}$ of C–H stretching showed the presence of decanoic acid on the surface of MNPs (after extraction) [30]. The morphology of MNPs was observed through the FESEM image (figure 2b).

## 3.2. Optimization of magnetic nanoparticles assisted dispersive liquid–liquid microextraction conditions

### 3.2.1. Effect of decanoic acid amount

Decanoic acid is an amphiphile and thus can conveniently form self-assembled aggregates in an aqueous sample [31]. The coacervation of decanoic acid and CAP in a conducive sample solution will lead to

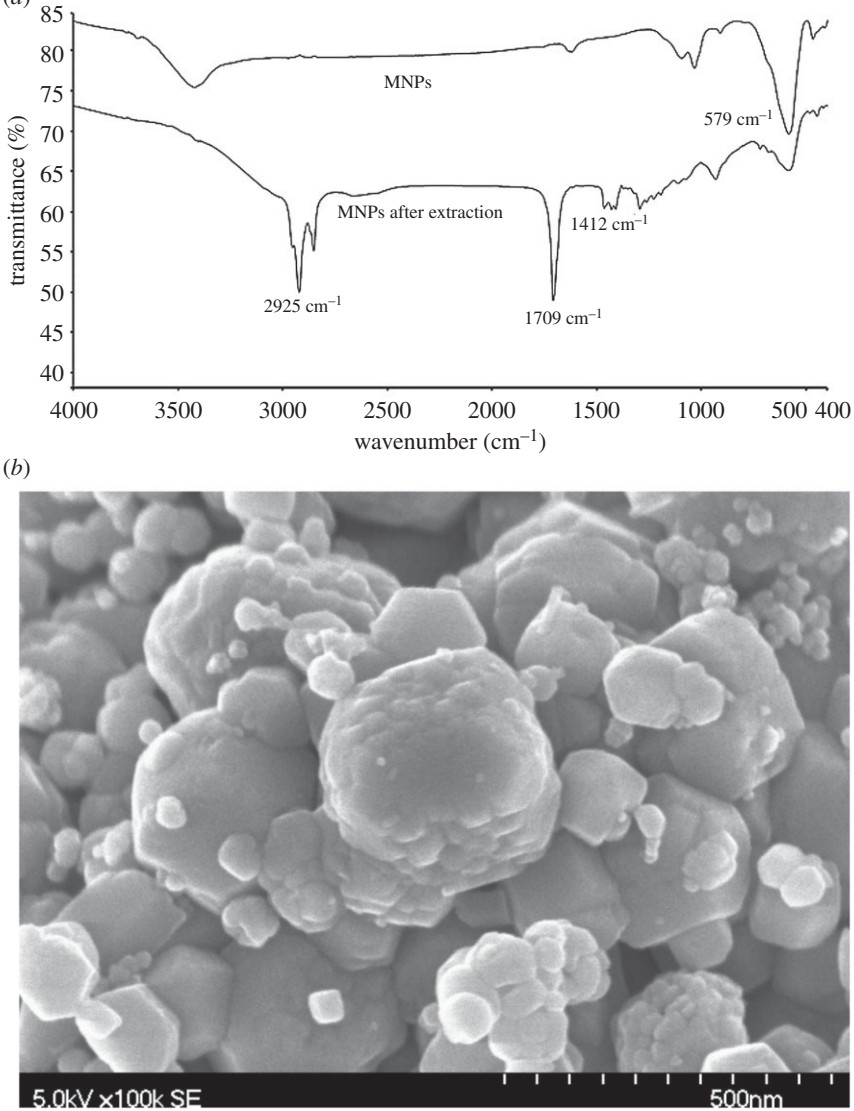

**Figure 2.** (*a*) FTIR spectra of MNPs and (*b*) FESEM image of MNPs.

micelle formation, forming vesicles entrapping CAP at the micellar core. This allows extraction of CAP from water samples with decanoic acid acting as the extraction solvent. Therefore, determining the optimum amount of decanoic acid to be used is crucial for maximum extraction efficiency of CAP. Amounts of decanoic acid ranging from 0 to 400 mg (w/v) were tested in the optimization study. The absorbance signal first increased as the decanoic acid amount increased, and it peaked at 200 mg (w/v). Subsequently, any additional amount of decanoic acid slightly reduced the extraction efficiency (figure 3*a*). The trend of increasing extraction efficiency up to 200 mg (w/v) of decanoic acid probably was owing to the increased coacervation fraction between decanoic acid and the target analytes [32]. However, the reduced extractability of CAP at amounts of decanoic acid above 200 mg (w/v) probably was owing to the higher viscosity of the solution, which would hinder the mass transfer process of CAP from sample solution to the extraction solvent. Thus, 200 mg (w/v) of decanoic acid was selected as the optimum amount for use in subsequent experiments.

### 3.2.2. Effect of extraction time

Extraction time is defined as the contact time between the aqueous solvent and the tested analyte. Generally, extraction efficiency is time dependent and defined by the time required for distribution equilibrium to occur between the target analytes and the aqueous solution [33]. In this study, the effect of incubation extraction times ranging from 3 to 20 min were tested while the other variables

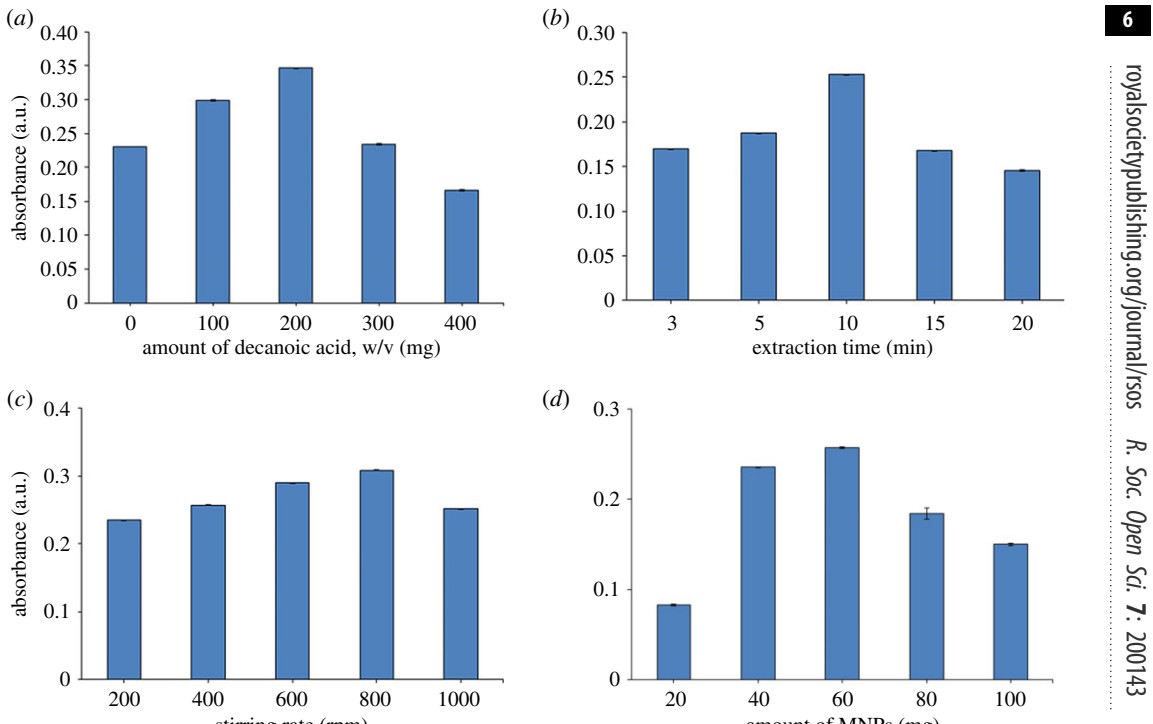

**Figure 3.** Effect of (*a*) amount of decanoic acid, (*b*) extraction time, (*c*) stirring rate, and (*d*) amount of MNPs on efficiency of DLLME-MNPs from spiked water (*n* = 3 in each case).

were kept constant. The extraction efficiency was time dependent (figure 3*b*). Extraction efficiency improved gradually at the earlier time range (3–10 min), peaked at 10 min and decreased thereafter. The improved extraction efficiency within a short time period (less than 10 min) suggests that efficient contact between the solvent and target analytes occurred [34], whereas the extraction efficiency slightly dropped when the extraction time was increased further. This phenomenon might be owing to the degradation of the extracted CAP after prolonge exposure [35]. Based on these results, 10 min of extraction time was chosen as the optimal condition for further experiments.

### 3.2.3. Effect of stirring rate

Stirring rate normally correlates positively with extraction efficiency, as it facilitates the mass transfer of the targeted analytes and the extracting material [36,37]. Stirring rates of 200, 400, 600, 800 and 1000 rpm were tested to determine the optimum rate for maximum extraction efficiency of CAP. Extraction efficiency increased with increasing stirring rate, reached its maximum value at 800 rpm and decreased thereafter (figure 3*c*). This trend is consistent with the results from previous reports, which suggested that stirring promotes proper contact area distribution [36] and increased penetration of mass transfer of the solutes [29]. The decreased extraction efficiency evident at a stirring rate of 1000 rpm may have been owing to the formation of a vortex, which could have resulted in poor contact between the analyte and the extracting solvent [36]. Therefore, a stirring rate of 800 rpm was selected as the optimum condition.

### 3.2.4. Effect of magnetic nanoparticles amount

MNPs were used as adsorbents for the separation of extracted solvent containing CAP from the aqueous matrices. This process eliminates the requirement for centrifugation or filtration and increases the yield of phase separation [31,38]. Direct contact between MNPs and decanoic acid allows the MNPs to adsorb CAP onto their surfaces. Different amounts of MNPs were tested (20–100 mg). The anticipated positive correlation between increasing amount of MNPs and extraction efficiency was evident up to 60 mg of MNPs, but the extraction efficiency decreased beyond this point (figure 3*d*). MNPs improve extraction efficiency because their nano-size provides a large surface area-to-volume ratio and shortens the diffusion route of the analytes [10,39]. Hydrogen bonding was likely to be formed and have an

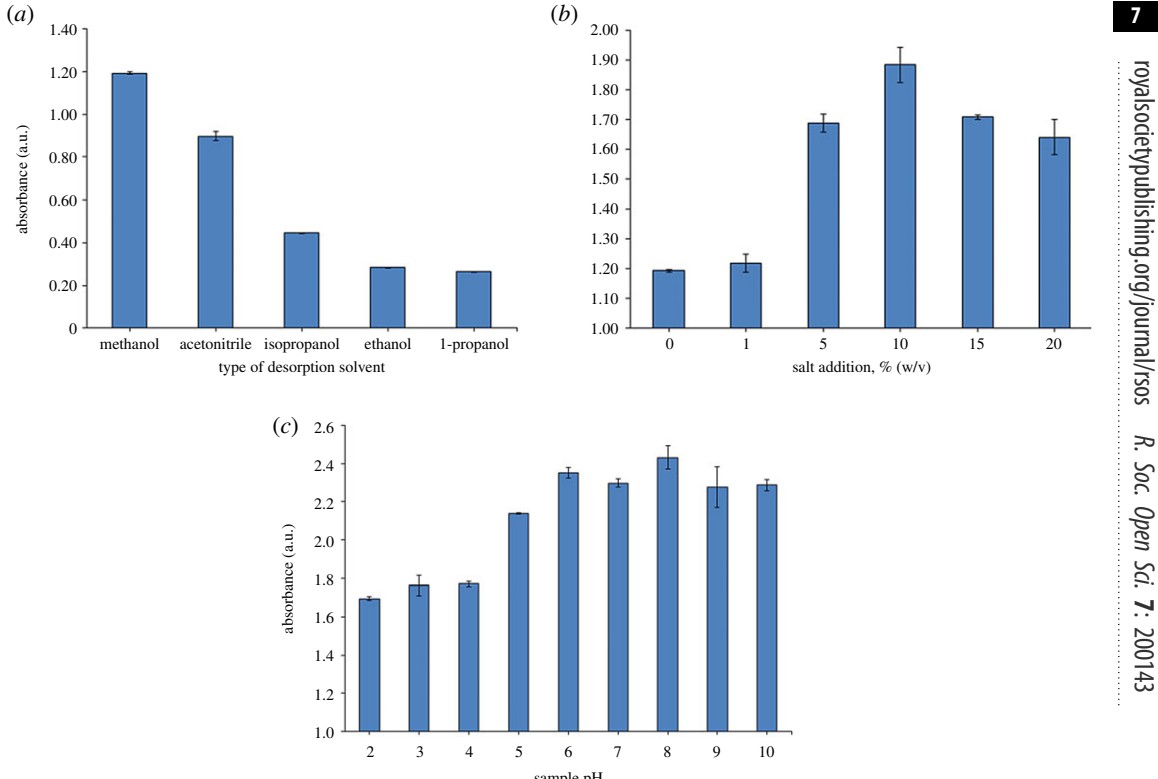

**Figure 4.** Effect of (a) type of desorption solvent, (b) salt addition, and (c) sample pH on efficiency of DLLME-MNPs from spiked water ($n = 3$ in each case).

important role in the adsorption mechanism between the MNPs and extraction solvent. Beyond 60 mg of MNPs, the extraction efficiency decreased, as the larger adsorbent amount to desorption volume ratio weakened the elution efficiency at the specific time and volume [40]. Based on these results, the optimum amount of MNPs was 60 mg.

### 3.2.5. Effect of desorption solvent

Desorption in extraction is a process whereby analytes are released from the surface of the adsorbent, thus it is important to use a suitable desorption solvent. Methanol, acetonitrile, isopropanol, ethanol and 1-propanol were evaluated as candidate desorption solvents, primarily because of their solubility with CAP and their compatibility with the UV–Vis spectrometer. Different polarity of solvents may affect the interaction between the solvent and solutes. Because CAP is relatively polar, a polar desorption solvent should produce better DLLME-MNPs extraction efficiency than less polar solvents. In this study, methanol was the most effective solvent for the desorption of CAP from the surface of MNPs, followed by acetonitrile, isopropanol, ethanol and 1-propanol (figure 4a). Therefore, methanol was selected as the optimum desorption solvent for use in the subsequent experiments.

### 3.2.6. Effect of salt addition

The addition of salt may cause an increase in an aqueous solution's ionic strength, which would then reduce the solubility of the target analyte in the sample solution and consequently enhance the extraction efficiency. In this study, the effect of salt addition was investigated by varying the amount of NaCl added (0–20%, w/v). Figure 4b clearly shows the positive correlation between CAP extraction recovery and increasing NaCl concentration in the range of 1 to 10% (w/v) and the downward trend of extraction recovery beyond 10% (w/v) NaCl addition. The maximum extraction efficiency was observed at 10% (w/v) of NaCl. The addition of salt increased the ionic strength of the aqueous phase, thereby enhancing the portion of CAP in the organic phase [32]. At values beyond 10% (w/v) NaCl, the salt concentration became saturated and affected the stability of the fine droplets. This resulted in high

**Table 1.** Validation of the DLLME-MNPs method validation of measuring CAP in water samples.

| analytical figures | |
|---|---|
| linear range (µg l$^{-1}$) | 50–1000 µg l$^{-1}$ |
| regression equation | $y = 0.0005x + 2.0444$ |
| $R^2$ | 0.9933 |
| LOD | 16.5 µg l$^{-1}$ |
| LOQ | 50.0 µg l$^{-1}$ |

**Table 2.** Measurement of CAP in water samples. (n.d. = not detected.)

| | tap water | | | lake water | | |
|---|---|---|---|---|---|---|
| spiked (µg l$^{-1}$) | found (µg l$^{-1}$) | recovery (%) | RSD (%) | found (µg l$^{-1}$) | recovery (%) | RSD (%) |
| 0 | n.d. | — | — | n.d | — | — |
| 50 | 46.4 | 92.7 | 0.69 | 45.5 | 91.0 | 2.93 |
| 200 | 184.4 | 92.2 | 0.45 | 183.2 | 91.6 | 6.29 |

viscosity of the solution, which impeded the transfer of CAP from the aqueous solution to the extractant [31,41]. Based on these results, 10% (w/v) NaCl was used in the subsequent experiment.

### 3.2.7. Effect of pH

Sample pH is key for determining the chemical states of analytes. Based on previously reported work [4], the most suitable sample pH for CAP determination was in the range of 6.0–10.0. CAP is expected to be decomposed and could not exist in molecular state when strong acid or alkaline condition was used. The extraction of analytes in their molecular or neutral form is expected to be easier than their ionized form [42]. In this study, the effect of pH on the extraction of CAP was investigated over the range of pH 2–10. It was found that the best extraction efficiency was obtained at pH 8 which gave the highest absorbance of CAP (figure 4c). Therefore, pH 8 was selected and used in further analysis.

## 3.3. Method validation and analytical performances

The final optimum DLLME-MNPs conditions were: amount of decanoic acid, 200 mg; extraction time, 10 min; stirring rate, 800 rpm; amount of MNPs, 60 mg; desorption solvent, methanol; NaCl addition, 10%; and sample pH, 8. The obtained linearity, regression equations, coefficient of determination ($R^2$), LOD and LOQ are listed in table 1. The method exhibited good linearity ($R^2 = 0.9933$) within the concentration range of 50 to 1000 µg l$^{-1}$. The LOD was 16.5 µg l$^{-1}$ and the LOQ was 50.0 µg l$^{-1}$. Repeatability or precision was expressed as RSD (%). The results showed good RSD values ranging from 0.45 to 6.29% ($n = 3$) for spiked CAP levels at 50 and 200 µg l$^{-1}$ (table 2).

## 3.4. Analysis of real samples

The proposed method was applied to measure CAP in tap water and lake water samples. Both samples showed negative results for CAP. Thus, to assess the applicability of the developed DLLME-MNPs to real environmental samples, tap water and lake water samples were spiked at two concentration levels of CAP (50 and 200 µg l$^{-1}$). Three replicate sample extractions were conducted for each concentration. The results showed satisfactory average recoveries ranging from 91.0 to 92.7% (table 3). The newly developed method was compared with other previously reported extraction methods for CAP from aqueous matrices [12,39,43,44] (table 3). The extraction time of DLLME-MNPs was relatively short compared to that of other methods because the microdroplets of extraction solvent (decanoic acid) accelerated the mass transfer process of analytes from the aqueous sample to the MNPs.

**Table 3.** Comparison of results of this study with those of other reported methods for measurement of CAP in aqueous matrices. (Abbreviations: SPME: solid phase microextraction; DLLME-MNPs: magnetic nanoparticles assisted dispersive liquid–liquid microextraction of chloramphenicol in water samples; SPE-HLLME: solid phase extraction-homogeneous liquid–liquid microextraction; QuEChERS: quick, easy, cheap, effective, rugged and safe.)

| sample preparation | instrument | LOD (µg l⁻¹) | LOQ (µg l⁻¹) | sample/concentration (µg l⁻¹) | RSD (%) | extraction time (min) | ref |
|---|---|---|---|---|---|---|---|
| SPME | LC | 0.1 | 0.3 | tap water | | 20 | [43] |
| | | | | 0.5 | 6.0 | | |
| | | | | 5 | 5.9 | | |
| | | | | 10 | 5.1 | | |
| | | 0.3 | 0.7 | seawater | | | |
| | | | | 1 | 5.7 | | |
| | | | | 10 | 5.5 | | |
| | | | | 30 | 5.4 | | |
| modified QuEChERS | LC-MS/MS | 0.045 | 0.1 | honey | | — | [44] |
| | | | | 0.15 | 0.51 | | |
| | | | | 1.5 | 1.05 | | |
| | | | | 15 | 0.64 | | |
| | | | | milk | | | |
| | | | | 0.15 | 1.96 | | |
| | | | | 1.5 | 0.83 | | |
| | | | | 15 | 0.39 | | |
| DLLME | HPLC | 12.5 | 37.5 | milk | | 15 | [12] |
| | | | | 150 | 11.3 | | |
| | | | | 300 | 2.4 | | |
| | | | | 600 | 3.1 | | |
| SPE-HLLME | HPLC | 0.1 | 0.5 | feed water | | 30 | [39] |
| | | | | 40 | 2.5 | | |
| | | | | drinking water | | | |
| | | | | 40 | 1.7 | | |
| DLLME-MNPs | UV-Vis | 16.5 | 50 | tap water | | 10 | this method |
| | | | | 50 | 0.69 | | |
| | | | | 200 | 0.45 | | |
| | | | | lake water | | | |
| | | | | 50 | 2.93 | | |
| | | | | 200 | 6.29 | | |

The repeatability in terms of RSDs for the DLLME-MNPs were comparable or even better than those of other methods. Although other methods offer better sensitivity (LOD and LOQ), DLLME-MNPs requires only a simple apparatus and instrumentation to measure CAP present at trace levels. DLLME-MNPs offers the remarkable advantages of requiring a minimum amount of MNPs adsorbent (60 mg) and organic solvent (1 ml), a simple and rapid extraction procedure, and sufficient sensitivity for CAP.

## 4. Conclusion

In the present study, a new extraction method (DLLME-MNPs) was developed for the extraction of CAP from water samples prior to UV–Vis spectrophotometry detection. From the real sample analysis, no

CAP was detected in any of the samples. The DLLME-MNPs technique was optimized and validated, and it offers several remarkable advantages over conventional methods, such as rapid and simple analysis and faster extraction time with good linearity, detection limit, repeatability, and extraction recovery. In addition, the developed DLLME-MNPs requires only a small amount of solvent (1 ml) and sorbent (60 mg) for each extraction. DLLME-MNPs is a simple, convenient, efficient and promising microscale extraction method for CAP analysis.

Data accessibility. All the datasets supporting this article are available at the Dryad Digital Repository (https://doi.org/10.5061/dryad.hdr7sqvdp) [45].

Authors' contributions. N.A.A. carried out all of the laboratory work, ran the data analysis; S.M.S. and N.Y. participated in the data analysis, design of the study, and helped in drafting the manuscript; M.M., M.S., N.N.M.Z., S.K., M.R., N.S.M.H. and W.N.W.I. provided advice about the study design and data analysis. All authors gave final approval for publication.

Competing interests. All authors declare that there is no conflict of interest.

Funding. This work was supported by the Ministry of Education Malaysia (Fundamental Research Grant Scheme–203/CIPPT/6711630) and Universiti Sains Malaysia (Short-Term–304/CIPPT/6315101, Research University Grant-1001/CIPPT/8011052). The authors also acknowledge Universiti Sains Malaysia for providing a scholarship (USM Fellowship 2/18) to S.M.S.

Acknowledgements. The authors thank the editors and anonymous reviewers for their helpful suggestions for this manuscript.

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
