## [Reviewer comments · Royal Society Open Science]

Review History

RSOS-190484.R0 (Original submission)

Review form: Reviewer 1

Is the manuscript scientifically sound in its present form?

Yes

Are the interpretations and conclusions justified by the results?

Yes

Is the language acceptable?

Yes

Is it clear how to access all supporting data?

Yes

Do you have any ethical concerns with this paper?

No

Have you any concerns about statistical analyses in this paper?

No

Recommendation?

Reject

Comments to the Author(s)

This paper demonstrates that the extraction of CAP using SUPRAS-MNP can be considered as simple, convenient, efficient and promising microscale extraction method of CAP analysis.

Although the paper is well organized, however, there are some major problems still need to be addressed.

1. The paper didn't provide the sufficient evidence to explain the reason of use decanoic acid as a supramolecular solvent in this method, and if there are other options?

2. The main components of magnetic nanoparticles materials were not showed in the paper , and is it possible to re-use the material?

3. This paper is missing of introduction to what is MNPs materials, including that directly related to the preparation and characterization of materials.

4. According to Table 3, the LOD and LQD of the method are too high compared with other reported methods for the determination of CAP in aqueous matrices, and failure to demonstrate the advantages of this method.

After careful evaluation of this manuscript, I don't think this manuscript suitable for publication in Royal Society Open Science before solve the above problems, because the manuscript does not offer a sufficient advance over previous works to meet the impact requirements of the journal.

Review form: Reviewer 2

Is the manuscript scientifically sound in its present form?

Yes

Are the interpretations and conclusions justified by the results?

Yes

Is the language acceptable?

Yes

Is it clear how to access all supporting data?

Yes

Do you have any ethical concerns with this paper?

No

Have you any concerns about statistical analyses in this paper?

No

Recommendation?

Accept with minor revision (please list in comments)

Comments to the Author(s)

The scientific results disseminated through the manuscript looks new and significant, however, minor revisions are needed in the text of the manuscript.

It's good if authors can include the acceptable level or limit of CAP in the environment (treated waste water, lake water etc.) set by the authorized bodies in the introduction.

Page 2, Line 13

Optimized parameters were as follows:

The word optimized should be changed to optimum.

Page 2, Line 15/16

Suggest to change ...good linearity... to ...acceptable linearity...

Page 2, Line 52

Remove the word "the" in ...(in **the** mg or μ g range)...

Page 3, Line 68 to 71

These sentences should be moved to previous paragraph that discuss on MNPs. The concluding remarks already supported the relationship between MNPs paragraph and SUPRAS paragraph, thus no need to explain on MNPs again here.

Page 4, Line 140

Change the sub-title to: Effect of decanoic acid amount

Page 5, Line 177

Change the sub-title to: Effect of MNPs amount

Page 5, Line 187

Change ...optimized amount... to ...optimum amount...

Page 5, Sub-title 3.1.7

Authors have mentioned about the present of analytes in the form of anionic, cationic or neutral at varied pH but did not relate this with the optimum pH chosen and its effect(s) on the analyte-acceptor diffusion ratio. Also mention what happen at pH 7.

Page 6, Line 231

Check on R2

Review form: Reviewer 3

Is the manuscript scientifically sound in its present form?

Yes

Are the interpretations and conclusions justified by the results?

Yes

Is the language acceptable?

Yes

Is it clear how to access all supporting data?

Not Applicable

Do you have any ethical concerns with this paper?

No

Have you any concerns about statistical analyses in this paper?

No

Recommendation?

Major revision is needed (please make suggestions in comments)

Comments to the Author(s)

The authors developed an extraction method termed magnetic nanoparticle assisted supramolecular solvent extraction for chloramphenicol (CAP). The extracted CAP was determined by UV-Vis spectrophotometry. The conditions for extraction of CAP were optimized in detail. Before the manuscript could be accepted by Royal Society Open Science, some issues should be addressed. The details are as follows.

1. Please provide the selectivity of the method for CAP. Do other antibiotics or substances in lake water interfere with the determination of CAP?

2. Please provide the chemical structure of CAP?

3. The MNPs were purchased from a company. Please provide their IR spectrum and zeta potential at different pHs. What are the moieties on the surface of MNPs? Why do the MNPs can adsorb the vesicles containing CAP?

4. Lines 180-181, the authors indicated ".....allows the MNPs to adsorb the positively charged vesicles onto their surface." The pKa of decanoic acid is 4.9. At pH 8.0, the vesicles containing decanoic acid should be negatively charged. Please measure their zeta potential.

5. "3.1.2. Effect of extraction time",the degradation of CAP may explain the lower extraction efficiency after prolonged time exposure. "3.1.3. Effect of stirring rate",the decreased extraction efficiency evident at stirring rate of 1000 rpm maybe due to the formation of vortex, which could have caused the poor contact between the analyte and the extracting solvent. Extraction time and stirring rate influencing the extraction efficiency could be the cross factors that influence each other.

Review form: Reviewer 4

Is the manuscript scientifically sound in its present form?

No

Are the interpretations and conclusions justified by the results?

No

Is the language acceptable?

Yes

Is it clear how to access all supporting data?

Yes

Do you have any ethical concerns with this paper?

No

Have you any concerns about statistical analyses in this paper?

Yes

Recommendation?

Major revision is needed (please make suggestions in comments)

Comments to the Author(s)

Recommendation: Publish after major revisions.

Comments:

Comments on manuscript: "Rapid magnetic nanoparticle assisted supramolecular solvent extraction for UV-Vis determination of chloramphenicol in water samples".

The manuscript describes the extraction method termed magnetic nanoparticle assisted supramolecular solvent extraction was developed for the extraction of chloramphenicol antibiotic from water samples prior to UV-Vis spectrophotometry detection. The manuscript may be accepted after a major revision. The followings are my comments and questions on the manuscript:

Some critical points in the paper are:

Phase separation can be achieved by self-assembly during supramolecular solvent extraction, why did the authors use MNPs?

What's the absolute extraction efficiency and recovery of the analytes during the extraction? How did the authors to make sure all of the micelles and vesicles were absorbed and separate from the bulk solution by MNPs?

Is it possible to entirely elute the formed supramolecular solvents from the surface of MNPs by only 20 μ L methanol? How did the authors to quantify the volume of the final mixture compositing of methanol and eluted solvent? It is not easy to do that. Please give more clear explanation on the procedure.

Could the authors demonstrate the possible mechanism of the adsorption of solvent with MNPs? The nanostructures of MNPs need to be characterized by SEM.

I am not happy with the acronym "SUPRAS". It is not a good acronym for this kind of extraction. In fact SUPRAS are not a "solvent" but consist of micelles or vesicles, which is not obvious from the term "supramolecular solvent = SUPRAS".

Review form: Reviewer 5

Is the manuscript scientifically sound in its present form?

No

Are the interpretations and conclusions justified by the results?

Yes

Is the language acceptable?

Yes

Is it clear how to access all supporting data?

Not Applicable

Do you have any ethical concerns with this paper?

No

Have you any concerns about statistical analyses in this paper?

Yes

Recommendation?

Reject

Comments to the Author(s)

RSOS-190484

In the manuscript entitled "Rapid magnetic nanoparticle assisted supramolecular solvent extraction for UV-Vis determination of chloramphenicol in water samples" chloramphenicol (CAP) as an antibiotic were extracted based on a magnetic nanoparticle-assisted supramolecular solvent extraction (SUPRAS-MNP) from water. Here decanoic acid was used as supramolecular

solvent. The coated MNPs were then desorbed with methanol and the clean extract was submitted to UV-Vis spectrophotometry detection. Some influencing parameters such as amount of decanoic acid, extraction time, stirring rate, amount of MNPs, type of desorption solvent, salt addition, and sample pH were optimized. Under the optimized conditions, figures of merit of the developed method were determined and the method was applied to the tap and lake water samples.

There are some issues regarding this manuscript listed as follows:

1. There are no error bars for reported values in some figures such as Fig. 2, Fig. 3 and Fig. 4 while in some others this issue has been considered. It should be implemented in whole figures.
2. The scale unit of MNPs amount is mg while in Fig.2 and in some places in the manuscript the (w/v) term is inserted instead of mg. It should be removed.
3. Page 5, line 163 (section 3.1.2); For justification of the signal dropping with the increase of extraction time, authors suggest the degradation of analyte and refer to TrAC- Trend Anal. Chem. 19, 229-248. (doi:10.1016/S0165-9936(99)00185-5), while there is no relevant content regarding the degradation of CAP, particularly. If in the mentioned reference, there is any related issue authors should point it out.
4. Why some other influencing parameters such as desorption solvent volume along with desorption time and temperature were not considered for optimization?
5. Page 7, line 233 (section 3.2); LOD value which is reported in this section ($15.6 \mu\text{g L}^{-1}$) is different with the values reported in Table 1 and abstract ($16.5 \mu\text{g L}^{-1}$). It should be replaced and checked.
6. Intercept and slope values, which are reported in Table 1, at the concentration range of $50\text{-}1000 \mu\text{g L}^{-1}$, result negative values, which is confirmed that, these values are false.

As a result, this manuscript has insufficient novelty and transparent analytical data under current status.

Decision letter (RSOS-190484.R0)

24-Apr-2019

Dear Dr Yahaya:

Manuscript ID: RSOS-190484

Title: "Rapid magnetic nanoparticle assisted supramolecular solvent extraction for UV-Vis determination of chloramphenicol in water samples"

Thank you for submitting the above manuscript to Royal Society Open Science. Your paper was sent to reviewers and their comments are included at the bottom of this letter.

In view of the concerns raised by the reviewers, the manuscript has been rejected in its current form. However, a new manuscript may be submitted which takes into consideration these comments.

Please note that resubmitting your manuscript does not guarantee eventual acceptance, and that your resubmission will be subject to peer review before a decision is made.

Once you have revised your manuscript, go to <https://mc.manuscriptcentral.com/rsos> and login to your Author Center. Click on "Manuscripts with Decisions," and then click on "Create a

Resubmission" located next to the manuscript number. Then, follow the steps for resubmitting your manuscript.

Your resubmitted manuscript should be submitted by 22-Oct-2019. If you are unable to submit by this date please contact the Editorial Office.

On behalf of the Subject Editor Professor Anthony Stace and the Associate Editor Dr Ya-Wen Wang

REVIEWER(S) REPORTS:

Associate Editor Comments to Author ():

RSC Associate Editor:

Comments to the Author:

(There are no comments.)

RSC Subject Editor:

Comments to the Author:

(There are no comments.)

Reviewers' Comments to Author:

Reviewer: 1

Comments to the Author(s)

This paper demonstrates that the extraction of CAP using SUPRAS-MNP can be considered as simple, convenient, efficient and promising microscale extraction method of CAP analysis.

Although the paper is well organized, however, there are some major problems still need to be addressed.

1. The paper didn't provide the sufficient evidence to explain the reason of use decanoic acid as a supramolecular solvent in this method, and if there are other options?
2. The main components of magnetic nanoparticles materials were not showed in the paper , and is it possible to re-use the material?
3. This paper is missing of introduction to what is MNPs materials, including that directly related to the preparation and characterization of materials.
4. According to Table 3, the LOD and LQD of the method are too high compared with other reported methods for the determination of CAP in aqueous matrices, and failure to demonstrate the advantages of this method.

After careful evaluation of this manuscript, I don't think this manuscript suitable for publication in Royal Society Open Science before solve the above problems, because the manuscript does not offer a sufficient advance over previous works to meet the impact requirements of the journal.

Reviewer: 2

Comments to the Author(s)

The scientific results disseminated through the manuscript looks new and significant, however, minor revisions are needed in the text of the manuscript.

Page 2, Introduction

It's good if authors can include the acceptable level or limit of CAP in the environment (treated waste water, lake water etc.) set by the authorized bodies in the introduction.

Page 2, Line 13

Optimized parameters were as follows:

The word optimized should be changed to optimum.

Page 2, Line 15/16

Suggest to change ...good linearity... to ...acceptable linearity...

Page 2, Line 52

Remove the word "the" in ...(in $\mu\text{g/L}$; mg or μg range)...

Page 3, Line 68 to 71

These sentences should be moved to previous paragraph that discuss on MNPs. The concluding remarks already supported the relationship between MNPs paragraph and SUPRAS paragraph, thus no need to explain on MNPs again here.

Page 4, Line 140

Change the sub-title to: Effect of decanoic acid amount

Page 5, Line 177

Change the sub-title to: Effect of MNPs amount

Page 5, Line 187

Change ...optimized amount... to ...optimum amount...

Page 5, Sub-title 3.1.7

Authors have mentioned about the present of analytes in the form of anionic, cationic or neutral at varied pH but did not relate this with the optimum pH chosen and its effect(s) on the analyte-acceptor diffusion ratio. Also mention what happen at pH 7.

Page 6, Line 231

Check on R2

Reviewer: 3

Comments to the Author(s)

The authors developed an extraction method termed magnetic nanoparticle assisted supramolecular solvent extraction for chloramphenicol (CAP). The extracted CAP was determined by UV-Vis spectrophotometry. The conditions for extraction of CAP were optimized in detail. Before the manuscript could be accepted by Royal Society Open Science, some issues should be addressed. The details are as follows.

1. Please provide the selectivity of the method for CAP. Do other antibiotics or substances in lake water interfere with the determination of CAP?
2. Please provide the chemical structure of CAP?

3. The MNPs were purchased from a company. Please provide their IR spectrum and zeta potential at different pHs. What are the moieties on the surface of MNPs? Why do the MNPs can adsorb the vesicles containing CAP?
4. Lines 180-181, the authors indicated ".....allows the MNPs to adsorb the positively charged vesicles onto their surface." The pKa of decanoic acid is 4.9. At pH 8.0, the vesicles containing decanoic acid should be negatively charged. Please measure their zeta potential.
5. "3.1.2. Effect of extraction time",the degradation of CAP may explain the lower extraction efficiency after prolonged time exposure. "3.1.3. Effect of stirring rate",the decreased extraction efficiency evident at stirring rate of 1000 rpm maybe due to the formation of vortex, which could have caused the poor contact between the analyte and the extracting solvent. Extraction time and stirring rate influencing the extraction efficiency could be the cross factors that influence each other.

Reviewer: 4

Comments to the Author(s)

Recommendation: Publish after major revisions.

Comments:

Comments on manuscript: "Rapid magnetic nanoparticle assisted supramolecular solvent extraction for UV-Vis determination of chloramphenicol in water samples".

The manuscript describes the extraction method termed magnetic nanoparticle assisted supramolecular solvent extraction was developed for the extraction of chloramphenicol antibiotic from water samples prior to UV-Vis spectrophotometry detection. The manuscript may be accepted after a major revision. The followings are my comments and questions on the manuscript:

Some critical points in the paper are:

Phase separation can be achieved by self-assembly during supramolecular solvent extraction, why did the authors use MNPs?

What's the absolute extraction efficiency and recovery of the analytes during the extraction? How did the authors to make sure all of the micelles and vesicles were absorbed and separate from the bulk solution by MNPs?

Is it possible to entirely elute the formed supramolecular solvents from the surface of MNPs by only 20 μ L methanol? How did the authors to quantify the volume of the final mixture compositing of methanol and eluted solvent? It is not easy to do that. Please give more clear explanation on the procedure.

Could the authors demonstrate the possible mechanism of the adsorption of solvent with MNPs? The nanostructures of MNPs need to be characterized by SEM.

I am not happy with the acronym "SUPRAS". It is not a good acronym for this kind of extraction. In fact SUPRAS are not a "solvent" but consist of micelles or vesicles, which is not obvious from the term "supramolecular solvent = SUPRAS".

Reviewer: 5

Comments to the Author(s)

RSOS-190484

In the manuscript entitled "Rapid magnetic nanoparticle assisted supramolecular solvent extraction for UV-Vis determination of chloramphenicol in water samples" chloramphenicol (CAP) as an antibiotic were extracted based on a magnetic nanoparticle-assisted supramolecular solvent extraction (SUPRAS-MNP) from water. Here decanoic acid was used as supramolecular solvent. The coated MNPs were then desorbed with methanol and the clean extract was submitted to UV-Vis spectrophotometry detection. Some influencing parameters such as amount of decanoic acid, extraction time, stirring rate, amount of MNPs, type of desorption solvent, salt addition, and sample pH were optimized. Under the optimized conditions, figures of merit of the

developed method were determined and the method was applied to the tap and lake water samples.

There are some issues regarding this manuscript listed as follows:

1. There are no error bars for reported values in some figures such as Fig. 2, Fig. 3 and Fig. 4 while in some others this issue has been considered. It should be implemented in whole figures.
2. The scale unit of MNPs amount is mg while in Fig.2 and in some places in the manuscript the (w/v) term is inserted instead of mg. It should be removed.
3. Page 5, line 163 (section 3.1.2); For justification of the signal dropping with the increase of extraction time, authors suggest the degradation of analyte and refer to TrAC- Trend Anal. Chem. 19, 229-248. (doi:10.1016/S0165-9936(99)00185-5), while there is no relevant content regarding the degradation of CAP, particularly. If in the mentioned reference, there is any related issue authors should point it out.
4. Why some other influencing parameters such as desorption solvent volume along with desorption time and temperature were not considered for optimization?
5. Page 7, line 233 (section 3.2); LOD value which is reported in this section (15.6 $\mu\text{g L}^{-1}$) is different with the values reported in Table 1 and abstract (16.5 $\mu\text{g L}^{-1}$). It should be replaced and checked.
6. Intercept and slope values, which are reported in Table 1, at the concentration range of 50-1000 $\mu\text{g L}^{-1}$, result negative values, which is confirmed that, these values are false.

As a result, this manuscript has insufficient novelty and transparent analytical data under current status.

Author's Response to Decision Letter for (RSOS-190484.R0)

See Appendix A.

RSOS-200143.R0

Review form: Reviewer 2

Is the manuscript scientifically sound in its present form?

Yes

Are the interpretations and conclusions justified by the results?

Yes

Is the language acceptable?

Yes

Do you have any ethical concerns with this paper?

No

Have you any concerns about statistical analyses in this paper?

No

Recommendation?

Accept as is

Comments to the Author(s)

All the corrections required have been taken into account.

Review form: Reviewer 3

Is the manuscript scientifically sound in its present form?

Yes

Are the interpretations and conclusions justified by the results?

Yes

Is the language acceptable?

Yes

Do you have any ethical concerns with this paper?

No

Have you any concerns about statistical analyses in this paper?

No

Recommendation?

Accept with minor revision (please list in comments)

Comments to the Author(s)

The authors have addressed the major issues raised by the Editor and Reviewers. Before the manuscript could be accepted by Royal Society Open Science, some minor issues should be addressed. The details are as follows.

1. I suggest the title to be "Magnetic nanoparticles assisted dispersive liquid-liquid microextraction of chloramphenicol in water samples". CAP was first extracted by decanoic acid, then the CAP-decanoic acid complex was adsorbed by magnetic nanoparticles. After CAP was eluted by methanol, decanoic acid was likely to be adsorbed by magnetic nanoparticles. Thus, the authors can use FT-IR or other characterization methods to prove whether decanoic acid is still on the magnetic nanoparticles after elution of CAP.

Review form: Reviewer 4

Is the manuscript scientifically sound in its present form?

Yes

Are the interpretations and conclusions justified by the results?

Yes

Is the language acceptable?

Yes

Do you have any ethical concerns with this paper?

No

Have you any concerns about statistical analyses in this paper?

No

Recommendation?

Accept as is

Comments to the Author(s)

The paper is revised well.

Review form: Reviewer 5**Is the manuscript scientifically sound in its present form?**

No

Are the interpretations and conclusions justified by the results?

No

Is the language acceptable?

Yes

Do you have any ethical concerns with this paper?

No

Have you any concerns about statistical analyses in this paper?

No

Recommendation?

Reject

Comments to the Author(s)

In the manuscript entitled "Dispersive liquid-liquid microextraction with extractant removal by magnetic nanoparticles for chloramphenicol preconcentration in water samples" a DLLME-MSPE-UV-Vis method for isolation and determination of chloramphenicol (CAP) antibiotic from water is developed. In this work, decanoic acid was used as DLLME solvent. The effect and amount of influencing parameters such as amount of decanoic acid, extraction time, stirring rate, amount of MNPs, desorption solvent type, salt addition, and sample pH were investigated and optimized. Under the optimum conditions, analytical data of the developed method were achieved and the method was applied to the tap and lake water samples.

There are some issues regarding this manuscript listed as follows:

In extraction time optimization section, to justify the signal dropping with the extraction time increase, authors are referring to the degradation of analyte as a reason and cite "TrAC- Trend Anal. Chem. 19, 229-248. (doi:10.1016/S0165-9936(99)00185-5)", while this reference has no relevant data on this regard, particularly for CAP. In the manuscript there is no data about the optimization of important influencing parameters such as desorption solvent volume along with the desorption time and temperature.

Considering the very large intercept value (2.044) rather than small slope value (0.0005) of the suggested line equation in Table 1 at the concentration range of 50-1000 $\mu\text{g L}^{-1}$, the developed method has a significant matrix and background effect, while the LOD and LQD values are too high compared with other reported methods. There are some relevant works which is not cited in the manuscript while they have developed sample preparation methods based on DLLME-SPE for isolation and determination of drugs such as CAP in food staff and aquatic matrices (Food Analytical Methods, 11, 2018, 759-767. DOI: 10.1007/s12161-017-1048-2 & Analytical and Bioanalytical Chemistry, 408, 2016, 1701-1713. DOI: 10.1007/s00216-015-9284-z).

Since the manuscript has no significant superiority over previous works and the lack of novelty remains an issue, this manuscript is not acceptable.

Decision letter (RSOS-200143.R0)

28-Feb-2020

Dear Dr Yahaya:

Title: Dispersive liquid-liquid microextraction with extractant removal by magnetic nanoparticles for chloramphenicol preconcentration in water samples
Manuscript ID: RSOS-200143

The editor assigned to your paper has now received comments from reviewers. We would like you to revise your paper in accordance with the referee and Subject Editor suggestions which can be found below (not including confidential reports to the Editor). Please note this decision does not guarantee eventual acceptance.

Please submit a copy of your revised paper before 22-Mar-2020. Please note that the revision deadline will expire at 00.00am on this date. If we do not hear from you within this time then it will be assumed that the paper has been withdrawn. In exceptional circumstances, extensions may be possible if agreed with the Editorial Office in advance. We do not allow multiple rounds of revision so we urge you to make every effort to fully address all of the comments at this stage. If deemed necessary by the Editors, your manuscript will be sent back to one or more of the original reviewers for assessment. If the original reviewers are not available we may invite new reviewers.

RSC Associate Editor
Comments to the Author:
(There are no comments.)

Reviewers' Comments to Author:
Reviewer: 3

Comments to the Author(s)

The authors have addressed the major issues raised by the Editor and Reviewers. Before the manuscript could be accepted by Royal Society Open Science, some minor issues should be addressed. The details are as follows.

1. I suggest the title to be "Magnetic nanoparticles assisted dispersive liquid-liquid microextraction of chloramphenicol in water samples". CAP was first extracted by decanoic acid, then the CAP-decanoic acid complex was adsorbed by magnetic nanoparticles. After CAP was eluted by methanol, decanoic acid was likely to be adsorbed by magnetic nanoparticles. Thus, the authors can use FT-IR or other characterization methods to prove whether decanoic acid is still on the magnetic nanoparticles after elution of CAP.

Reviewer: 2

Comments to the Author(s)
All the corrections required have been taken into account.

Reviewer: 4

Comments to the Author(s)
The paper is revised well.

Reviewer: 5

Comments to the Author(s)

In the manuscript entitled "Dispersive liquid-liquid microextraction with extractant removal by magnetic nanoparticles for chloramphenicol preconcentration in water samples" a DLLME-MSPE-UV-Vis method for isolation and determination of chloramphenicol (CAP) antibiotic from water is developed. In this work, decanoic acid was used as DLLME solvent. The effect and amount of influencing parameters such as amount of decanoic acid, extraction time, stirring rate, amount of MNPs, desorption solvent type, salt addition, and sample pH were investigated and optimized. Under the optimum conditions, analytical data of the developed method were achieved and the method was applied to the tap and lake water samples.

There are some issues regarding this manuscript listed as follows:

In extraction time optimization section, to justify the signal dropping with the extraction time increase, authors are referring to the degradation of analyte as a reason and cite "TrAC- Trend Anal. Chem. 19, 229-248. (doi:10.1016/S0165-9936(99)00185-5)", while this reference has no relevant data on this regard, particularly for CAP. In the manuscript there is no data about the

optimization of important influencing parameters such as desorption solvent volume along with the desorption time and temperature.

Considering the very large intercept value (2.044) rather than small slope value (0.0005) of the suggested line equation in Table 1 at the concentration range of 50-1000 $\mu\text{g L}^{-1}$, the developed method has a significant matrix and background effect, while the LOD and LQD values are too high compared with other reported methods. There are some relevant works which is not cited in the manuscript while they have developed sample preparation methods based on DLLME-SPE for isolation and determination of drugs such as CAP in food staff and aquatic matrices (Food Analytical Methods, 11, 2018, 759-767. DOI: 10.1007/s12161-017-1048-2 & Analytical and Bioanalytical Chemistry, 408, 2016, 1701-1713. DOI: 10.1007/s00216-015-9284-z).

Since the manuscript has no significant superiority over previous works and the lack of novelty remains an issue, this manuscript is not acceptable.

Author's Response to Decision Letter for (RSOS-200143.R0)

See Appendix B.

Decision letter (RSOS-200143.R1)

13-Mar-2020

Dear Dr Yahaya:

Title: Magnetic nanoparticles assisted dispersive liquid-liquid microextraction of chloramphenicol in water samples

Manuscript ID: RSOS-200143.R1

It is a pleasure to accept your manuscript in its current form for publication in Royal Society Open Science. The chemistry content of Royal Society Open Science is published in collaboration with the Royal Society of Chemistry.

RSC Associate Editor
Comments to the Author:
(There are no comments.)

Reviewer(s)' Comments to Author:

Appendix A

Response to editor and reviewers' comments

Date: 24th January 2020

Manuscript ID: RSOS-190484

Title: "Rapid magnetic nanoparticle assisted supramolecular solvent extraction for UV-Vis determination of chloramphenicol in water samples"

Dear Dr Laura Smith,

Editor,

Royal Society Open Science,

Thank you very much for organizing the review of our manuscript (RSOS-190484). We would like to thank the reviewers for their careful reading and thoughtful comments of our manuscript. We have considered all the comments given by the reviewers and revised the manuscript accordingly (The changes are highlighted). Based on the reviewers' comments, we have revised the manuscript title to: **Dispersive liquid-liquid microextraction with extractant removal by magnetic nanoparticles for chloramphenicol preconcentration in water samples.**

We believe that we have addressed all the points raised by the reviewers. We have gone through proofreading service for this manuscript and the certificate is attached together with the revised version of manuscript. We hope that the manuscript is now ready for consideration in Royal Society Open Science.

Sincerely yours,

Noorfatimah Yahaya, Ph.D. (Corresponding author)

Senior Lecturer,

Integrative Medicine Cluster,

Advanced Medical and Dental Institute,

Universiti Sains Malaysia,

Bertam,

13200 Kepala Batas,

Penang, Malaysia.

Tel: +604-5622071

Email: noorfatimah@usm.my

Reviewers' comments and our response

Reviewer: 1

Comments to the Author(s):

This paper demonstrates that the extraction of CAP using SUPRAS-MNP can be considered as simple, convenient, efficient and promising microscale extraction method of CAP analysis. Although the paper is well organized, however, there are some major problems still need to be addressed.

1.The paper didn't provide the sufficient evidence to explain the reason of use decanoic acid as a supramolecular solvent in this method, and if there are other options?

Response: Dear Reviewer #1, thank you very much for your valuable comments. We have revised the manuscript's title to 'Dispersive liquid-liquid microextraction with extractant removal by magnetic nanoparticles for chloramphenicol preconcentration in water samples.' In order to implement your comments, the role of decanoic acid as extraction solvent has been included in text. The changes are highlighted. Please see the introduction section (Lines 94-95).

2. The main components of magnetic nanoparticles materials were not showed in the paper , and is it possible to re-use the material?

Response: We thank the reviewer for the comment. In actual fact, the magnetic nanoparticles used in this work is commercially sourced from the Bendosen Laboratory Chemicals (Bendosen, Norway). Since only 60 mg was used for each extraction, the materials were discarded after single use to avoid any carry over effects in the experiments.

3.This paper is missing of introduction to what is MNPs materials, including that directly related to the preparation and characterization of materials.

Response: We appreciate the suggestion of reviewer and changes have been made accordingly. Introduction on MNPs has been included in text (Lines 88-91). The characterization of FTIR and VSM have been cited in text and we have included FESEM image as well (Fig. 1).

4.According to Table 3, the LOD and LQD of the method are too high compared with other reported methods for the determination of CAP in aqueous matrices, and failure to demonstrate the advantages of this method.

After careful evaluation of this manuscript, I don't think this manuscript suitable for publication in Royal Society Open Science before solve the above problems, because the manuscript does not offer a sufficient advance over previous works to meet the impact requirements of the journal.

Response: We appreciate the opinion of reviewer. However, in our opinion, the developed DLLME-MNPs method avoids tedious operation in extraction, providing an interesting and innovative approach of combining microscale sample preparation and magnetic adsorbent, DLLME and MNPs procedures. This approach has an 'environmentally-friendly' in nature due to its minimum amount of adsorbent consumption in each extraction. And more importantly, the procedure of conducting DLLME-MNPs is simple, easy and involved unsophisticated equipment (UV-Vis spectrophotometry, vortex and hotplate stirrer), which generally be found in a typical analytical laboratory. We hope the reviewer can consider accepting our manuscript.

Reviewer: 2

Comments to the Author(s):

The scientific results disseminated through the manuscript looks new and significant, however, minor revisions are needed in the text of the manuscript.

1. Page 2, Introduction

It's good if authors can include the acceptable level or limit of CAP in the environment (treated waste water, lake water etc.) set by the authorized bodies in the introduction.

Response: Dear Reviewer #2, thank you very much for your valuable comments. We appreciate the suggestion of reviewer. To the best of our knowledge, no regulation has been established for CAP in water samples. Thus, sensitive and reliable detection techniques are crucial to regularly monitor the residual CAP present in environmental waters and to assist in the establishment of the maximum residue limits for CAP. The information has been included in text (Lines 56-58).

2. Page 2, Line 13

Optimized parameters were as follows:

The word optimized should be changed to optimum.

Response: Optimized has been changed to optimum (Line 34).

3. Page 2, Line 15/16

Suggest to change ...good linearity... to ...acceptable linearity...

Response: good linearity has been changed to acceptable linearity (Line 36)

4. Page 2, Line 52

Remove the word "the" in ...(in **the** mg or μ g range)...

Response: The sentence has been omitted after revision (Change in title and methodology's term)

5. Page 3, Line 68 to 71

These sentences should be moved to previous paragraph that discuss on MNPs. The concluding remarks already supported the relationship between MNPs paragraph and SUPRAS paragraph, thus no need to explain on MNPs again here.

Response: The methodology has been revised based on the reviewer comments from SUPRAS-MNPs to DLLME-MNPs. Thus, the original paragraph has been deleted.

6. Page 4, Line 140

Change the sub-title to: Effect of decanoic acid amount

Response: Changes have been made accordingly (Line 166).

7. Page 5, Line 177

Change the sub-title to: Effect of MNPs amount

Response: Changes have been made accordingly (Line 204).

8. Page 5, Line 187

Change ...optimized amount... to ...optimum amount...

Response: Correction has been made accordingly (Line 215).

9. Page 5, Sub-title 3.1.7

Authors have mentioned about the present of analytes in the form of anionic, cationic or neutral at varied pH but did not relate this with the optimum pH chosen and its effect(s) on the analyte-acceptor diffusion ratio. Also mention what happen at pH 7.

Response: Based on previously reported work [39], the most suitable sample pH for CAP determination was in the range of 6.0–10.0. CAP is expected to be decomposed and could not exist in molecular state when strong acid or alkaline condition was used. The extraction of analytes in their molecular or neutral form is expected to be easier than their ionized form [40]. In this study, the effect of pH on the extraction of CAP was investigated over the range of pH 2 to 10. It was found that the best extraction efficiency was obtained at pH 8 which gave the highest absorbance of CAP (Fig. 3c). Therefore, pH 8.0 was selected and used in further analysis. Changes have been made accordingly (Lines 242-248).

10. Page 6, Line 231

Check on R2

Response: Corrections have been made accordingly (Line 254).

Reviewer: 3

Comments to the Author(s)

The authors developed an extraction method termed magnetic nanoparticle assisted supramolecular solvent extraction for chloramphenicol (CAP). The extracted CAP was determined by UV-Vis spectrophotometry. The conditions for extraction of CAP were optimized in detail. Before the manuscript could be accepted by Royal Society Open Science, some issues should be addressed. The details are as follows.

1. Please provide the selectivity of the method for CAP. Do other antibiotics or substances in lake water interfere with the determination of CAP?

Response: Dear Reviewer #3, thank you very much for your valuable comments. Based on our experimental data, we achieved acceptable precision (repeatability – <7%) and recovery (accuracy – 91-91.6%) of the clean extract, thus, we believe this procedure is selective and reliable towards the extraction of CAP.

2. Please provide the chemical structure of CAP?

Response: Chemical structure of CAP has been included in Fig. 2.

3. The MNPs were purchased from a company. Please provide their IR spectrum and zeta potential at different pHs. What are the moieties on the surface of MNPs? Why do the MNPs can adsorb the vesicles containing CAP?

Response: We thank the reviewer for the comments. The IR spectrum and Vibrating-sample magnetometer analysis of the MNPs have been reported by our group. A reference has been included in the text (Lines 112-113). We apologise for the mistakes. The paragraph on optimization has been revised accordingly. Based on

previously reported work [39], the most suitable sample pH for CAP determination was in the range of 6.0–10.0. CAP is expected to be decomposed and could not exist in molecular state when strong acid or alkaline condition was used. The extraction of analytes in their molecular or neutral form is expected to be easier than their ionized form [40]. In this study, the effect of pH on the extraction of CAP was investigated over the range of pH 2 to 10. It was found that the best extraction efficiency was obtained at pH 8 which gave the highest absorbance of CAP (Fig. 3c). Therefore, pH 8.0 was selected and used in further analysis. Changes have been made accordingly (Lines 238-245).

4. Lines 180-181, the authors indicated “.....allows the MNPs to adsorb the positively charged vesicles onto their surface.” The pKa of decanoic acid is 4.9. At pH 8.0, the vesicles containing decanoic acid should be negatively charged. Please measure their zeta potential.

Response: We thank the reviewer for the suggestion. We have corrected our statement and revised the sentence accordingly. (Please see Lines 242-248, Effect of pH).

5. “3.1.2. Effect of extraction time”,the degradation of CAP may explain the lower extraction efficiency after prolonged time exposure. “3.1.3. Effect of stirring rate”,the decreased extraction efficiency evident at stirring rate of 1000 rpm maybe due to the formation of vortex, which could have caused the poor contact between the analyte and the extracting solvent. Extraction time and stirring rate influencing the extraction efficiency could be the cross factors that influence each other.

Response: Thank you for your valuable comments. In this work, the main parameters affecting the extraction of CAP were optimized and investigated by one variable at-a-time and other parameters were kept constant. Considering the acceptable sensitivity, recovery and repeatability values, we think that the parameters used in this study are appropriate to evaluate the method performance. We appreciate the suggestion of reviewer and may consider the suggestion to study the cross factors that influence the extraction time and stirring rate of the developed method in our future study.

Reviewer: 4

Comments to the Author(s)

Recommendation: Publish after major revisions.

Comments:

Comments on manuscript: "Rapid magnetic nanoparticle assisted supramolecular solvent extraction for UV-Vis determination of chloramphenicol in water samples".

The manuscript describes the extraction method termed magnetic nanoparticle assisted supramolecular solvent extraction was developed for the extraction of chloramphenicol antibiotic from water samples prior to UV-Vis spectrophotometry detection. The manuscript may be accepted after a major revision. The followings are my comments and questions on the manuscript:

Some critical points in the paper are:

1. Phase separation can be achieved by self-assembly during supramolecular solvent extraction, why did the authors use MNPs?

Response: Dear Reviewer #4, thank you very much for your valuable comments. We have revised the manuscript's title to 'Dispersive liquid-liquid microextraction with extractant removal by magnetic nanoparticles for chloramphenicol preconcentration in water samples.' In order to implement your comments, we proposed the use of MNPs to simplify and speed up the recovery of procedure. Similar strategies have been used by other researchers [27] (Lines 88-91).

2. What's the absolute extraction efficiency and recovery of the analytes during the extraction? How did the authors to make sure all of the micelles and vesicles were absorbed and separate from the bulk solution by MNPs?

Response: We thank the reviewer for the suggestion. The recoveries for method validation were relative recoveries, which were calculated by comparing the extracted amount of CAP from water sample with the corresponding spiking amount on the calibration curve. The quantification of CAP was made based on the obtained recovery values in the range of 91–91.6%. Considering the acceptable sensitivity, recoveries

and repeatability of the developed DLLME-MNPs method using UV-Vis determination, we believe that this method has demonstrated good performance towards the extraction of CAP molecule.

3. Is it possible to entirely elute the formed supramolecular solvents from the surface of MNPs by only 20 μL methanol? How did the authors to quantify the volume of the final mixture compositing of methanol and eluted solvent? It is not easy to do that. Please give more clear explanation on the procedure.

Response: We thank the reviewer pointing this out. We have revised the manuscript's title to 'Dispersive liquid-liquid microextraction with extractant removal by magnetic nanoparticles for chloramphenicol preconcentration in water samples.' Indeed, decanoic acid used in this study was employed as an extraction solvent to extract CAP. We have corrected the term 'supramolecular solvent' to 'extraction solvent' throughout the manuscript.

After extraction, the MNPs were collected using an external magnetic field and the supernatant was discarded and 150 μL of methanol were added to desorb CAP from the MNPs surface. 150 μL was chosen because it is the minimum amount that can be used to completely immersed the MNPs. The mixture was sonicated for 2 min. An external magnet was applied, and the eluent was collected. The collected eluent was diluted with 850 μL of methanol and 500 μL of ultrapure water. Finally, the extracted samples were analysed using UV-Vis spectrophotometry. Based on our experimental data, we achieved acceptable precision (repeatability) and recovery (accuracy) of the clean extract, thus, we believe this procedure is consistent throughout the study (Lines 143-147).

4. Could the authors demonstrate the possible mechanism of the adsorption of solvent with MNPs?

Response: We appreciate the suggestion of reviewer and a sentence on mechanism of adsorption of extraction solvent with MNPs has been included in text (Lines 211-

213). Hydrogen bonding was likely to be formed and have an important role in the adsorption mechanism between the MNPs and extraction solvent.

5. The nanostructures of MNPs need to be characterized by SEM.

Response: A FESEM image of MNPs has been included in text (Fig. 1).

6. I am not happy with the acronym "SUPRAS". It is not a good acronym for this kind of extraction. In fact SUPRAS are not a "solvent" but consist of micelles or vesicles, which is not obvious from the term "supramolecular solvent = SUPRAS".

Response: We appreciate and agree with the comment of the reviewer. We apologize for the mistakes. The manuscript has been revised and the title has been changed to 'Dispersive liquid-liquid microextraction with extractant removal by magnetic nanoparticles for chloramphenicol preconcentration in water samples'.

Reviewer: 5

Comments to the Author(s)

RSOS-190484

In the manuscript entitled "Rapid magnetic nanoparticle assisted supramolecular solvent extraction for UV-Vis determination of chloramphenicol in water samples" chloramphenicol (CAP) as an antibiotic were extracted based on a magnetic nanoparticle-assisted supramolecular solvent extraction (SUPRAS-MNP) from water. Here decanoic acid was used as supramolecular solvent. The coated MNPs were then desorbed with methanol and the clean extract was submitted to UV-Vis spectrophotometry detection. Some influencing parameters such as amount of decanoic acid, extraction time, stirring rate, amount of MNPs, type of desorption solvent, salt addition, and sample pH were optimized. Under the optimized conditions, figures of merit of the developed method were determined and the method was applied to the tap and lake water samples.

There are some issues regarding this manuscript listed as follows:

1. There are no error bars for reported values in some figures such as Fig. 2, Fig. 3 and Fig. 4 while in some others this issue has been considered. It should be implemented in whole figures.

Response: Dear Reviewer #5, thank you very much for your valuable comments. We have included error bars in all figures. Some error bars might not be seen since the values are too small (Fig.3-Fig.4).

2. The scale unit of MNPs amount is mg while in Fig.2 and in some places in the manuscript the (w/v) term is inserted instead of mg. It should be removed.

Response: We thank the reviewer for pointing this out. We apologize for the errors. Changes have been made accordingly (Fig. 3 and lines 205-215).

3. Page 5, line 163 (section 3.1.2); For justification of the signal dropping with the increase of extraction time, authors suggest the degradation of analyte and refer to TrAC- Trend Anal. Chem. 19, 229-248. (doi:10.1016/S0165-9936(99)00185-5), while there is no relevant content regarding the degradation of CAP, particularly. If in the mentioned reference, there is any related issue authors should point it out.

Response: We thank the reviewer for pointing this out. We apologise for the inaccurate information in this section. Corrections have been made accordingly (Lines 187-190).

3. Why some other influencing parameters such as desorption solvent volume along with desorption time and temperature were not considered for optimization?

Response: We appreciate and agree with the comment of the reviewer. However, this study is focusing on the development of new and rapid extraction technique and CAP was selected as model compounds in this work to study the feasibility of the developed method for the extraction of CAP from water samples. The developed method would be one of the alternatives that is easy to use, simple, required minimum amount of sorbent and rapid for the determination of organic compounds in the future. Based on our experimental data, we achieved acceptable precision (repeatability) and recovery

(accuracy) of the clean extract, thus, we believe this procedure is consistent throughout the study. We appreciate the suggestion of reviewer and may consider the suggestion in our future study.

4. Page 7, line 233 (section 3.2); LOD value which is reported in this section ($15.6 \mu\text{g L}^{-1}$) is different with the values reported in Table 1 and abstract ($16.5 \mu\text{g L}^{-1}$). It should be replaced and checked.

Response: We thank the reviewer for the comments. Corrections have been made accordingly (Table 1, abstract-line 37, line 255).

6. Intercept and slope values, which are reported in Table 1, at the concentration range of $50\text{-}1000 \mu\text{g L}^{-1}$, result negative values, which is confirmed that, these values are false.

As a result, this manuscript has insufficient novelty and transparent analytical data under current status.

Response: We appreciate the opinion of reviewer. Corrections have been made accordingly and the negative values have been corrected. In our opinion, the developed DLLME-MNPs method avoids tedious operation in extraction, providing an interesting and innovative approach of combining microscale sample preparation and magnetic adsorbent, DLLME and MNPs procedures. This approach has an 'environmentally-friendly' in nature due to its minimum amount of organic solvent and adsorbent consumption in each extraction. And more importantly, the procedure of conducting DLLME-MNPs is simple, easy and involved unsophisticated equipment (UV-Vis spectrophotometry, vortex and hotplate stirrer), which generally be found in a typical analytical laboratory. We hope the reviewer can consider accepting our manuscript.

Appendix B

Response to editor

Date: 4th of March 2020

Manuscript ID: RSOS-200143

Title: "Dispersive liquid-liquid microextraction with extractant removal by magnetic nanoparticles for chloramphenicol preconcentration in water samples"

Dear Dr Laura Smith,

Editor,

Royal Society Open Science,

Thank you very much for organizing the review of our manuscript (RSOS-200143). We would like to thank the reviewers for their careful reading and thoughtful comments of our manuscript. We have considered all the comments given by the reviewers and revised the manuscript accordingly (The changes are highlighted). Based on the reviewers' comments, we have revised the manuscript title to: Magnetic nanoparticles assisted dispersive liquid-liquid microextraction of chloramphenicol in water samples.

We believe that we have addressed all the points raised by the reviewers. We hope that the manuscript is now ready for publication in Royal Society Open Science.

Sincerely yours,

Noorfatimah Yahaya, Ph.D. (Corresponding author)

Senior Lecturer,

Integrative Medicine Cluster,

Advanced Medical and Dental Institute,

Universiti Sains Malaysia,

Bertam,

13200 Kepala Batas,

Penang, Malaysia.

Tel: +604-5622071

Email: noorfatimah@usm.my

Response to reviewers' comments

Reviewers' Comments to Author:

Reviewer: 3

Comments to the Author(s):

The authors have addressed the major issues raised by the Editor and Reviewers. Before the manuscript could be accepted by Royal Society Open Science, some minor issues should be addressed. The details are as follows.

1. I suggest the title to be "Magnetic nanoparticles assisted dispersive liquid-liquid microextraction of chloramphenicol in water samples". CAP was first extracted by decanoic acid, then the CAP-decanoic acid complex was adsorbed by magnetic nanoparticles. After CAP was eluted by methanol, decanoic acid was likely to be adsorbed by magnetic nanoparticles. Thus, the authors can use FT-IR or other characterization methods to prove whether decanoic acid is still on the magnetic nanoparticles after elution of CAP.

Response: Dear Reviewer #1, thank you very much for your valuable comments. We appreciate and agree with the suggestions of the reviewer. Title of the manuscript has been revised to 'Magnetic nanoparticles assisted dispersive liquid-liquid microextraction of chloramphenicol in water samples'.

FTIR results have been included in the manuscript with one added reference [30], to prove the presence of trace decanoic acid on the magnetic nanoparticles after elution of CAP. (Figure 1(a), Text: Lines 120-121, 164-169)

Reviewer: 2

Comments to the Author(s):

1. All the corrections required have been taken into account.

Response: We thank the reviewer for his positive comments on our work.

Reviewer: 4

Comments to the Author(s):

1. The paper is revised well.

Response: We thank the reviewer for his positive comments on our work.

Reviewer: 5

Comments to the Author(s):

In the manuscript entitled "Dispersive liquid-liquid microextraction with extractant removal by magnetic nanoparticles for chloramphenicol preconcentration in water samples" a DLLME-MSPE-UV-Vis method for isolation and determination of chloramphenicol (CAP) antibiotic from water is developed. In this work, decanoic acid was used as DLLME solvent. The effect

and amount of influencing parameters such as amount of decanoic acid, extraction time, stirring rate, amount of MNPs, desorption solvent type, salt addition, and sample pH were investigated and optimized. Under the optimum conditions, analytical data of the developed method were achieved and the method was applied to the tap and lake water samples.

There are some issues regarding this manuscript listed as follows:

1. In extraction time optimization section, to justify the signal dropping with the extraction time increase, authors are referring to the degradation of analyte as a reason and cite "TrAC-Trend Anal. Chem. 19, 229-248. (doi:10.1016/S0165-9936(99)00185-5)", while this reference has no relevant data on this regard, particularly for CAP. In the manuscript there is no data about the optimization of important influencing parameters such as desorption solvent volume along with the desorption time and temperature.

Response: Dear Reviewer #5, thank you very much for your valuable comments. We thank the reviewer for pointing this out. We apologise for the inaccurate reference in this section. Corrections have been made accordingly with one new added reference [35]. (Lines 196-197).

We appreciate the suggestions of reviewer. There is no doubt that desorption solvent volume, desorption time and temperature are important parameters that require optimization in the present study. For the desorption solvent volume, we used the minimum amount of desorption solvent that is required to completely filled the cuvette of UV spectrophotometer. The minimum amount of desorption solvent will produce the highest enrichment factor of analyte. Therefore, 1.5 mL was selected as the best desorption solvent volume in this study.

Based on previously reported work by Safari *et al.*, 2015 [31 from text], a method based on magnetic nanoparticle assisted supramolecular solvent extraction using decanoic acid as extraction solvent for triazine herbicides was developed. The optimum temperature used in the respective study was 36 °C for interaction of magnetic nanoparticles and decanoic acid. Therefore, by considering the similar extraction solvent with magnetic nanoparticles used in this work, 36 °C was used as optimum temperature.

For desorption time, several reported literatures investigated on dispersive liquid-liquid microextraction combined with magnetic nanoparticles techniques [1-3], the authors found that 2 min was sufficient to desorb target analytes from the magnetic nanoparticles' adsorbent. Some authors have mentioned that agglomeration of nanoparticles, and therefore trapping of the analytes in the interstices of the sorbent particles, could be responsible for the significant decrease after this time [3]. Thus, in our opinion, 2 min can be considered as the best desorption time for chloramphenicol antibiotic in this study.

References:

[31] Safari, M., Yamini, Y., Tahmasebi, E., & Ebrahimpour, B. (2016). Magnetic nanoparticle assisted supramolecular solvent extraction of triazine herbicides prior to their determination by HPLC with UV detection. *Microchimica Acta*, 183(1), 203–210. <https://doi.org/10.1007/s00604-015-1607-4>

[1] Sadeghi, S., & Oliaei, S. (2016). Optimization of ionic liquid based dispersive liquid-liquid microextraction combined with dispersive micro-solid phase extraction for the spectrofluorimetric determination of sulfasalazine in aqueous samples by response surface methodology. *RSC Advances*, 6(114), 113551–113560. <https://doi.org/10.1039/C6RA20223C>

[2] Zhao, J., Zhu, Y., Jiao, Y., Ning, J., & Yang, Y. (2016). Ionic-liquid-based dispersive liquid–liquid microextraction combined with magnetic solid-phase extraction for the determination of aflatoxins B1, B2, G1, and G2 in animal feeds by high-performance liquid chromatography with fluorescence detection. *Journal of Separation Science*, 39(19), 3789–3797. <https://doi.org/10.1002/jssc.201600671>

[3] Khalilian, F., Rezaee, M., & Gorgabi, M. K. (2015). Magnetic polypyrrole/Fe₃O₄ particles as an effective sorbent for the extraction of abamectin from fruit juices using magnetic solid-phase extraction combined with dispersive liquid-liquid microextraction. *Analytical Methods*, 7(5), 2182–2189. <https://doi.org/10.1039/c4ay03074e>

2. Considering the very large intercept value (2.044) rather than small slope value (0.0005) of the suggested line equation in Table 1 at the concentration range of 50-1000 µg L⁻¹, the developed method has a significant matrix and background effect, while the LOD and LQD values are too high compared with other reported methods. There are some relevant works which is not cited in the manuscript while they have developed sample preparation methods based on DLLME-SPE for isolation and determination of drugs such as CAP in food staff and aquatic matrices (*Food Analytical Methods*, 11, 2018, 759-767. DOI: 10.1007/s12161-017-1048-2 & *Analytical and Bioanalytical Chemistry*, 408, 2016, 1701-1713. DOI: 10.1007/s00216-015-9284-z).

Response: We appreciate the opinion of reviewer. In this study, we have evaluated the linearity, repeatability (precision) and recovery (accuracy) of the proposed method over the concentration range of 50-1000 µg L⁻¹. Based on the experimental data, acceptable coefficient of determination ($R^2 = 0.9933$), precision (repeatability <7% RSD) and recovery (91–91.6%) were obtained using water samples. Thus, we believe this procedure is reliable and applicable towards the extraction of chloramphenicol antibiotic from water samples. Although the obtained LOD and LOQ values using DLLME-MNPs were relatively high compared with other reported methods, it covers part per billion range of targeted analyte, which is in our opinion, a satisfactory range obtained by the UV-Vis spectrophotometry instrument.

Additionally, the developed DLLME-MNPs combined with UV-Vis method avoids tedious operation in extraction, providing a simple and rapid approach by combining microscale sample preparation and magnetic adsorbent, DLLME and MNPs procedures. The use of MNPs in this method has simplified the collection process. More importantly, the procedure of conducting DLLME-MNPs involved unsophisticated instrument such as UV-Vis spectrophotometry, vortex and hotplate stirrer, which generally be found in a typical analytical laboratory. In our humble point of view, the developed method readily lends itself as useful extraction tool for quantitative and qualitative determination of chloramphenicol in water samples within the concentration range of 50-1000 µg L⁻¹.

Relevant works based on similar sample preparation methods of chloramphenicol have been cited in text. (Ref [13] and [35])

3. Since the manuscript has no significant superiority over previous works and the lack of novelty remains an issue, this manuscript is not acceptable.

Response: We appreciate the opinion of reviewer. However, in our opinion, every method published possessed advantages and disadvantages. There are a few factors to be considered when chemists employ a method. The factors include method simplicity, availability of the equipment required, acceptable LODs, skill of the analysts etc. In our opinion, this approach has an added value by using decanoic acid as extraction solvent in dispersive liquid-liquid microextraction (DLLME) compared to more hazardous and toxic halogenated organic solvents used in the conventional version of DLLME. Additionally, the

used of MNPs in this study has significantly simplified the collection procedure. And more importantly, the procedure of conducting DLLME-MNPs is simple and can be carried out in any routine analytical laboratory. Thus, the use of decanoic acid as extraction solvent in DLLME, in combination with MNPs procedure could represent a new strategy for the extraction of chloramphenicol from water samples. We appreciate the comment of reviewer and will further improve the approach in our future study.